# Embed and Emulate: Learning to estimate parameters of dynamical systems with uncertainty quantification

**Ruoxi Jiang**
Department of Computer Science
University of Chicago
Chicago, IL 60637
roxie62@uchicago.edu

**Rebecca Willett**
Department of Statistics and Computer Science
University of Chicago
Chicago, IL 60637
willett@uchicago.edu

## Abstract

This paper explores learning emulators for parameter estimation with uncertainty estimation of high-dimensional dynamical systems. We assume access to a computationally complex simulator that inputs a candidate parameter and outputs a corresponding multichannel time series. Our task is to accurately estimate a range of likely values of the underlying parameters. Standard iterative approaches necessitate running the simulator many times, which is computationally prohibitive. This paper describes a novel framework for learning feature embeddings of observed dynamics jointly with an emulator that can replace high-cost simulators for parameter estimation. Leveraging a contrastive learning approach, our method exploits intrinsic data properties within and across parameter and trajectory domains. On a coupled 396-dimensional multiscale Lorenz 96 system, our method significantly outperforms a typical parameter estimation method based on predefined metrics and a classical numerical simulator, and with only 1.19% of the baseline's computation time. Ablation studies highlight the potential of explicitly designing learned emulators for parameter estimation by leveraging contrastive learning.

## 1 Introduction

Physics-based simulators play a vital role in many domains of science and engineering, from energy infrastructure to atmospheric sciences. They are frequently critical for assessing risk and exploring "what if" scenarios, which require running models many times [Snyder et al., 2019, Beusch et al., 2020, Helgeson et al., 2021, Zhao et al., 2021]. However, the increasing complexity of operational computer models makes running such simulators a major challenge. Emulators (also known as surrogate models) are models trained to mimic numerical simulations at a much lower computational cost, particularly for parameters or inputs that have not been simulated.

This paper considers the problem of estimating the parameters of a physics simulation that provide the best fit to data. The estimation problem can generally be thought of as a nonlinear inverse problem in which our goal is to estimate parameters $\phi \in \mathbb{R}^k$ from a noisy multichannel time series $\mathbf{Z} \in \mathbb{R}^{T \times d}$. $\mathbf{Z} = H(\phi; \mathbf{Z}_0) + \eta$, where $H(\phi; \mathbf{Z}_0)$ represents running is a physics simulator with parameters $\phi$ and initial condition $\mathbf{Z}_0$ for $T$ time steps, and $\eta \sim \mathcal{N}(0, \bar{\Gamma})$ is observation noise. To ease the notation, we drop the initial condition $\mathbf{Z}_0$ in $H(\phi; \mathbf{Z}_0)$. We are particularly interested in complex simulators for which we do not have analytic expressions for $H$, evaluating $H(\phi)$ is a computational bottleneck and we cannot readily compute its gradients.

One important application of this problem arises in climate science, where climate scientists have spent decades developing sophisticated physics-based models corresponding to $H$, often implemented using large-scale software systems that solve complex systems of differential equations [Kay et al., 2015]. In this case, evaluating $H$ can be computationally demanding. Furthermore, tools like physics-informed neural networks [Raissi et al., 2019] are inapplicable because we cannot compute

36th Conference on Neural Information Processing Systems (NeurIPS 2022).

losses that depend upon knowing the form of $H$. Estimating the parameters of such models based on observational data is essential for accurate climate forecasting, and we typically seek not only point estimates of parameters, but also quantifiable uncertainty measures that allow us to forecast the full range of possible future outcomes. Parameter uncertainty quantification is vital here [Cleary et al., 2021, Souza et al., 2020, Hansen, 2022], especially when dealing with noisy observations where a small error in $\hat{\phi}$ might lead to a dramatically different forecasts of the dynamics; uncertainty quantification also plays a vital role in the decision-making process to prevent hazardous loss under tail events [Hansen, 2022].

We explore an alternative framework in which we use our simulator to generate training samples of the form $(\phi_i, \mathbf{Z}_i)$ for $i = 1, \ldots, n$ and train a machine learning model for parameter estimation. Our approach, which we call EMBED & EMULATE, jointly trains two neural networks. The first, $f_\theta$, maps an observation $\mathbf{Z}$ to a low-dimensional embedding. The second, $\hat{g}_\theta$, emulates the map $g_\theta := f_\theta \circ H$, and so maps a parameter vector $\phi$ to the same low-dimensional embedding space. The learned $\hat{g}_\theta$ may be used in place of $H$ within an optimization-based framework such as Ensemble Kalman Inversion (EnKI, §2).

More specifically, our method uses the learned low-dimensional embedding to specify an objective function well-suited to parameter estimation without expert knowledge and the learned emulator as a surrogate for classical numerical methods used to compute $H$. Furthermore, inspired by empirical Bayes methods, our learned network suggests a natural mechanism for specifying the EnKI prior based on the trained network.

## 2 Related work

**Brief parameter estimation background:** A widely-used approach for parameter estimation is Ensemble Kalman Inversion (EnKI; §A.1) [Iglesias et al., 2013]. EnKI is a derivative-free optimization framework allows a user to specify a prior distribution over the parameter vector $\phi$ and results in samples from the corresponding posterior distribution. As a result, this method provides users with important information about the uncertainty of estimates of $\phi$. The clear and thought-provoking paper of Schneider et al. [2017] highlighted the use of the EnKI in earth system modeling.

Despite its success in the perfect-model setting (i.e., where $H(\phi)$ can be computed exactly and perfectly represents the dynamics in $\mathbf{Z}$ for some $\phi$), using such methods in practice presents several challenges. To use the EnKI for parameter estimation, we must specify three key elements: (a) the objective function that EnKI is attempting to minimize; (b) the tool used to compute the forward mapping $H$ (or a surrogate for $H$); and (c) a prior $p_\phi$. For instance, Schneider et al. [2017] uses a Gaussian prior $p_\phi$, computes $H$ using Runge-Kutta methods [Dormand and Prince, 1980], and attempts to minimize the Mahalanobis distance

$$J_{\text{moment}}(\phi) := \|m(\mathbf{Z}_i) - m(H(\phi))\|^2_{\boldsymbol{\Sigma}(m(\mathbf{Z}_i))}, \tag{1}$$

where $m(\mathbf{Z}_i)$ is a vector of first and second moments of different spatial channels of $\mathbf{Z}_i$ and $\boldsymbol{\Sigma}_i$ is a diagonal matrix with the $j$-th diagonal entry $\boldsymbol{\Sigma}_{i,j,j} := \text{Var}[m(H(\phi_i))_j]$. Using a loss based on moments within the EnKI framework provides critical robustness to the chaotic nature of $\mathbf{Z}_i$.

This framework presents two central challenges. First, choosing the objective for EnKI to minimize requires non-trivial domain knowledge, and a poor choice may lead to biased parameter estimates or unpredictable sensitivities to certain features in $\mathbf{Z}$. Second, high-resolution simulations used to evaluate $H$ for a given $\phi$ (e.g. Runge-Kutta or other classical numerical solvers) can be computationally demanding. In particular, the EnKI method uses a collection of *particles* to represent samples of the posterior, and the forward model $H$ must be calculated for each particle at each iteration. For complex optimization landscapes or in high-dimensional settings, the number of particles must be large. This is further complicated by the fact that often the parameter estimation task is conducted repeatedly, e.g., each time new climate data is acquired.

**Simulation-based inference (SBI):** A large body of work in SBI focuses on parameter estimation for physics-based simulators [Beaumont et al., 2009, Papamakarios et al., 2019, Cranmer et al., 2020, Lueckmann et al., 2019, Chen et al., 2020b, Alsing et al., 2019]. Advanced SBI methods focus on adaptive generation of new training data to approximate the posterior and could be classified into different categories based on how they adaptively choose informative simulations [Lueckmann et al., 2021]. One approach, likelihood estimation, proposes to approximate an intractable likelihood function [Drovandi et al., 2018]. For instance, Sequential Neural Likelihood estimation (SNL, Papamakarios et al. [2019]) trains a conditional neural density estimator (e.g., Masked Autoregressive

Flow (MAF)) which models the conditional distribution of data given parameters [Papamakarios et al., 2017]). While SNL does not readily scale to high-dimensional data, a recent variant called SNL+ [Chen et al., 2020b] addresses this limitation by learning sufficient statistics (embeddings) of the data based on the infomax principle, and it iteratively updates the network used to compute sufficient statistics and the neural density estimators during sequential sampling. An alternative approach, (sequential) neural posterior estimation ((S)NPE), aims to approximate directly the target posterior (SNPE-A in Papamakarios and Murray [2016], SNPE-B in Lueckmann et al. [2017], SNPE-C in Greenberg et al. [2019]), where SNPE-C forces fewer restrictions on the form of prior and posterior by leveraging neural conditional density estimation.

However, in the context of this manuscript's setting, i.e., estimating multiple $\phi_i$ for multiple different observations $\mathbf{Z}_i$ at test time, the key technique of SBI, sequential sampling, can require substantial computational investments. This idea is discussed further with our experimental results.

**Learned emulators:** Physics models and simulations are pervasive in weather and climate science, astrophysics, high-energy and accelerator physics, and the study of dynamical systems. These models and simulations are used, for example, to infer the underlying physical processes and equations that govern our observations, design new sensors or facilities, estimate errors, understand experimental behavior, and estimate unknown parameters within models. Many such physics simulators require expensive computational resources, and are difficult to fully leverage. "Learned emulators," "surrogate models," or "approximants" are computationally-efficient approximate models that use numerically simulated data to train a machine learning system to mimic these numerical simulations at a much smaller computational cost. There have been many recent successes in the development of surrogate model foundations [Brockherde et al., 2017, Raissi et al., 2019, Chattopadhyay et al., 2019, 2020, Raissi et al., 2017, Vlachas et al., 2018, 2020, Brajard et al., 2020, Gagne et al., 2020] and applications [Goel et al., 2008, Mengistu and Ghaly, 2008, Brigham and Aquino, 2007, Kim et al., 2015, Papadopoulos et al., 2018, White et al., 2019, Gentine et al., 2018, Rasp et al., 2018, Cohen et al., 2020, Yuval and O'Gorman, 2020, Xue et al., 2021, Wang et al., 2019].

Typically, one would directly try to emulate $H$, and the efficacy of the learned emulator would be evaluated on how well $\hat{H}(\phi)$ matches $H(\phi)$ (often using squared error) across a range of $\phi$. This formulation, however, may be suboptimal when the emulator will be used for parameter estimation. First, as we will detail later, emulating $H$ when $\mathbf{Z}$ is very high-dimensional is quite challenging. (In our experiments, $\dim(\mathbf{Z}) = 396,000$.) Furthermore, in many climate settings, $H$ represents a chaotic process in which very small changes to the initial conditions can result in very large differences later in the process. In this setting, training an emulator with a loss akin to $(1/n)\sum_{i=1}^{n} \|\mathbf{Z}_i - \hat{H}(\phi_i)\|_2^2$ may be overly sensitive to noise and initial conditions and not preserve statistical features of $\mathbf{Z}_i$ that may be essential to parameter estimation.

Inspired by the objective in (1), Cleary et al. [2021] trained a model $\hat{g}_\theta$ to emulate the moments of parameters using the loss function $\ell_{\mathrm{moment}}(\theta) := (1/n)\sum_{i=1}^{n} \|m(\mathbf{Z}_i) - \hat{g}_\theta(\phi_i)\|_{\mathbf{\Sigma}_i}^2$ to ease the burden of running $H$. However, $\ell_{\mathrm{moment}}(\theta)$ is not always the best choice. First, estimating $\mathbf{\Sigma}_i$ for each training sample is an enormous computational burden that far exceeds the cost of generating the original training data (unless the range of candidate $\phi$ is very small). Second, choosing the moment function $m$ (essentially fixing a particular low-dimensional embedding) is a critical design element that requires domain expertise and often must be tuned in practice. Finally, even if one is willing to generate accurate estimates of $\mathbf{\Sigma}_i$, one could face additional challenges. Cleary et al. [2021] is designed for inferring parameters given one fixed test observation, and they train their emulator for a small domain of parameter space dependent on the test observation at hand. Extending this framework to a larger domain of parameters $\phi$ results in large variability among the collection of corresponding $\mathbf{\Sigma}_i$s generated for training, making the loss landscape challenging to navigate.

**Contrastive learning:** With the above challenges in mind, we present a novel and generic framework for parameter estimation by jointly optimizing (a) an embedding of the multichannel dynamics used to evaluate the accuracy of a candidate parameter $\phi$ and (b) an emulator of the dynamics projected into the embedding space. We leverage a contrastive framework to learn discriminative representations for high-dimensional spatio-temporal data.

Contrastive representation learning has exploded in popularity recently for self-supervised visual feature learning that has achieved comparable performance with its supervised counterparts [Caron et al., 2020, Chen et al., 2020a, He et al., 2020, Grill et al., 2020, Zhang and Maire, 2020, Caron et al., 2021]. The majority of these frameworks operate under the push-pull principle for instance-wise

discrimination: images generated from different forms of data augmentation (e.g., cropping, color jittering) are pulled together while other images are pushed away. Apart from its common setup in unsupervised representation learning, Khosla et al. [2020] also demonstrated selecting positive and negative pairs in a supervised way leveraging label information can boost the performance. Many efforts focus on understanding the mechanism of contrastive learning [Oord et al., 2018, Wei et al., 2020, Arora et al., 2019, Wang and Isola, 2020, Zimmermann et al., 2021]. Among them, Zimmermann et al. [2021] show that contrastive loss can be interpreted as the cross-entropy between the latent conditional distribution and ground truth distribution.

## 3 Contributions

In our work, we follow the contrastive framework and learn an embedding network to capture structural information for high-dimensional spatio-temporal data and an emulator that can be used to compute parameter estimates with uncertainty estimates. This is effective for several reasons. First, when the model $H$ is bijective, representations learned using contrastive losses are highly correlated with their underlying parameters [Zimmermann et al., 2021]. Second, most existing emulating methods focus on approximating dynamical models for a fixed parameter $\phi$ and lack generalization capacity [Krishnapriyan et al., 2021]. In part this is due to the difficulties of emulating dynamics with high spatio-temporal resolution. We circumvent this bottleneck by training an emulator to output the learned latent representations of the dynamics, which are much lower-dimensional and easier to learn with fewer training samples. Finally, our method is inspired by the Contrastive Language–Image Pre-training (CLIP) [Radford et al., 2021] framework, which is designed for contrastive learning of aligned representations of images and language. Our approach, by analogy, learns aligned representations of dynamics and parameters, allowing us to jointly learn the embedding function and emulator for high-resolution data.

More specifically, this paper makes the following key contributions:

- Incorporation of a contrastive loss function for learning an embedding of the simulation outputs ($\mathbf{Z}_i$) instead of relying on a known "good" moment function $m$, leveraging ideas from CLIP [Radford et al., 2021];

- Development of an emulator designed to facilitate parameter estimation with uncertainty quantification. We demonstrate that the proposed emulator can be used effectively within an EnKI framework to generate parameter estimates with accurate posterior distribution estimates.

- An empirical evaluation of different methods (EnKI [Schneider et al., 2017], supervised learning, NPE-C Greenberg et al. [2019], SNL+ [Chen et al., 2020b], and our proposed emulator-based approach) in terms of robustness to noise or errors in training and testing data for a range of sample sizes as well as computational costs. Compared to using numerical solvers, we show that our method achieves higher accuracy overall in 1.08% of the computation time, even accounting for the computational costs of generating training data.

- Evaluation of the quality of the uncertainty estimates using the Continuous Ranked Probability Score (CRPS) for varying numbers of training samples.

Our empirical results highlight both the improved computational and empirical accuracy of our proposed emulator-centric approach relative to competing methods and, more broadly, the benefits of designing emulators specifically for parameter estimation tasks.

## 4 Proposed method

Our goal is to learn a mapping from observations of a multichannel dynamical system, $\mathbf{Z} \in \mathbb{R}^{T \times d}$, to the underlying system parameters $\phi \in \mathbb{R}^k$, where $\mathbf{Z} = H(\phi) + \eta$ for some noise vector $\eta \in \mathbb{R}^{T \times d}$. We assume we do not have access to an analytical expression for $H$, but can compute $H(\phi)$ for any $\phi$ using a computationally complex simulator. We further assume a prior $p_\phi$ over parameter space.

Using this prior and the simulator, we generate $n$ training samples as follows: for $i = 1, \ldots, n$, draw $\phi_i \sim p_\phi$, and use the simulator to compute $\mathbf{Z}_i = H(\phi_i) + \eta_i$. We do not explicitly generate noise $\eta_i$; rather, the numerical algorithm used to simulate $H$ will generally produce some errors, with faster implementations of $H$ being more error-prone. The distribution $p_\phi$ coupled with the noise distribution over $\eta$ induces the joint distribution $p_{\phi, \mathbf{z}}$.

### 4.1 Baseline approach – supervised learning

Given training samples $(\phi_i, \mathbf{Z}_i)$ for $i = 1, \ldots, n$, we train a neural network represented by $h_\theta$ using a ResNet [He et al., 2016] as the backbone (see details in §B.2). $\theta$ denotes the learned network parameters and our prediction is $\hat{\phi}_i = h_\theta(\mathbf{Z}_i)$. We train using the Mean Absolute Percentage Error (MAPE) loss

$$\ell_{\text{MAPE}}(\theta) := \frac{1}{n} \sum_{i=1}^{n} \sum_{j=1}^{k} \left| \frac{\phi_{ij} - (h_\theta(\mathbf{Z}_i))_j}{|\phi_{ij}| + \epsilon} \right|. \tag{2}$$

As we will see in the experimental results, this approach yields point estimates of $\phi$ on holdout test data with reasonable accuracy and offers a significant computational advantage over the EnKI method. However, it does not offer uncertainty estimates, and its test accuracy is lower than that of our proposed approach (§4.2).

### 4.2 Our approach: joint embedding and emulation via contrastive feature representation

**Contrastive feature representation:** Measuring the distance between a pair of dynamics, a necessary task for constructing a loss function used for training, is particularly challenging when the dynamics are chaotic. As mentioned above (§2), one alternative in the literature is to use a custom loss function based on summary statistics of $\mathbf{Z}$; this requires expert knowledge to determine which statistics are relevant, and has demonstrated efficacy only for narrow ranges of parameters $\phi$.

We present an alternative framework in which we learn an embedding of $\mathbf{Z}$, denoted $f_\theta(\mathbf{Z})$, that preserves statistical and structural characteristics of $\mathbf{Z}$ *most relevant to the parameter estimation task* and hence can be used to train an emulator. Here $\theta$ denotes all the parameters of the neural network defining the embedding function. Inspired by recent advances in contrastive representation learning, from a collection of parameters and trajectories pairs $\{\phi_i, \mathbf{Z}_i\}_{i=1}^{n}$, we propose a trajectory encoder $f_\theta$ which learns a distinguishable representation by minimizing the following variant of the Info Noise Contrastive Estimation (InfoNCE) loss [Oord et al., 2018]:

$$\ell_{\mathbf{ZZ}}(\theta; \tau) := \frac{1}{n} \sum_{i=1}^{n} -\log \frac{\exp\left(\langle f_\theta(\mathbf{Z}_i), f_\theta(\tilde{\mathbf{Z}}_i)\rangle/\tau\right)}{\sum\limits_{j=1}^{n} \exp(\langle f_\theta(\mathbf{Z}_i), f_\theta(\mathbf{Z}_j)\rangle/\tau)}. \tag{3}$$

Here $f_\theta(\mathbf{Z}) \in \mathbb{S}^{p-1}$ is normalized and lives in a unit hypersphere. $\tilde{\mathbf{Z}}_i$ is selected through data augmentation (see §B.1) or in a supervised way by measuring the distance in the parameter space: $\tilde{\mathbf{Z}}_i := \arg\min_{\mathbf{Z}_j \neq \mathbf{Z}_i} \delta(\mathbf{Z}_j, \mathbf{Z}_i)$, where $\delta$ is a metric used in calculating distance in parameter space, and $\tau$ is a temperature hyperparameter balancing the impact of similar pairs and negative examples [Zhang et al., 2018, Ravula et al., 2021]. In this way, we drive the model to embed data with similar values of the parameter $\phi$ in similar locations while repulsing data with unrelated parameters, resulting in an embedding that respects the latent structure of the parameter and trajectory pairs.

**Emulators:** Emulating the dynamics of a simulator by training a deep neural network $\hat{g}_\theta(\cdot)$ is a natural approach to easing the computational burden of parameter estimation. However, recent works in emulating dynamics focus on the fixed-parameter setting [Krishnapriyan et al., 2021] and cannot be easily generalized to multi-parameter settings (i.e. the emulator only approximates $H(\phi)$ for a single fixed $\phi$ instead of a range of $\phi$s, as we need for parameter estimation).

Learning emulators for a range of parameter values is challenging in part due to the dimension of the dynamics to be emulated (i.e. the dimension of the $\mathbf{Z}$s). To address this problem, we propose to learn an emulator $\hat{g}_\theta$ of $g_\theta := f_\theta \circ H$, which represents the mapping from the parameter space ($\phi$) to the latent representation space $f_\theta(\mathbf{Z})$. which is much lower-dimensional than $\mathbf{Z}$ and hence easier to learn with limited training samples.

We say that the trajectory encoder and the emulator are *perfectly aligned* when $\hat{g}_\theta(\phi) = f_\theta(\mathbf{Z}), \forall (\phi, \mathbf{Z}) \sim p_{\phi, \mathbf{Z}}$. Under perfect alignment, given a test $\mathbf{Z}$, we may seek to minimize

$$J(\phi; \mathbf{Z}, f_\theta, \hat{g}_\theta) = \|\hat{g}_\theta(\phi) - f_\theta(\mathbf{Z})\|_2^2 \tag{4}$$

instead of (1) with much faster objective function evaluations.

**Learning the emulator via CLIP:** Pretraining an embedding $f_\theta$ and then learning an emulator $\hat{g}_\theta$ with an $L_2$ loss is hard to optimize; the different architectures, necessitated by the very different dimensions of $\phi$ and $\mathbf{Z}$, make feature alignment challenging and the optimization prone to poor local

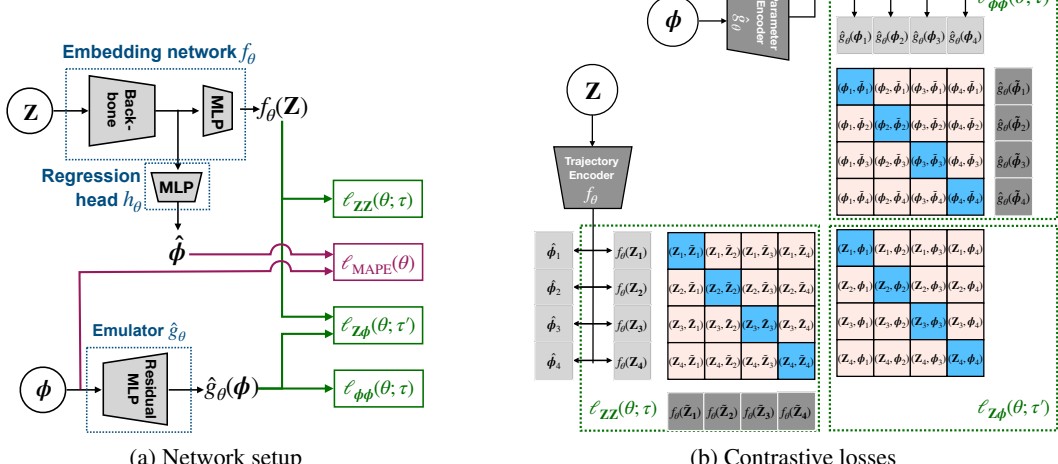

|                          |                          |
|--------------------------|--------------------------|
| (a) Network setup        | (b) Contrastive losses   |

Figure 1: **The** EMBED & EMULATE **framework.** (a) Components of our network and loss function. (b) We use three contrastive learning schemes. The bottom left block denotes the intra-trajecotry domain contrastive mechanism, the upper right denotes the intra-parameter domain contrastive learning, and the bottom right block shows inter-trajectory-parameter domains contrastive learning. Within each block, diagonals correspond to the dot product between the representations of positive trajectories pairs $(\mathbf{Z}_i, \tilde{\mathbf{Z}}_i)$, positive parameter pairs $(\phi_i, \tilde{\phi}_i)$, and matched trajectory parameter pairs $(\mathbf{Z}_i, \phi_i)$, which we aim to maximize. Off-diagonal pale pink blocks show similarities between the anchor point and the negatives examples, which we aim to minimize.

minimizers. Alternatively, inspired by the recent success of CLIP [Radford et al., 2021, Ravula et al., 2021] in cross-domain representation learning, we use the following variant of InfoNCE loss to align the representation space between two networks:

$$\ell_{\mathbf{Z}\phi}(\theta; \tau') := \frac{1}{n} \sum_{i=1}^{n} -\log \frac{\exp\left(\langle f_\theta(\mathbf{Z}_i), \hat{g}_\theta(\phi_i) \rangle / \tau'\right)}{\sum_{j=1}^{n} \exp\left(\langle f_\theta(\mathbf{Z}_i), \hat{g}_\theta(\phi_j) \rangle / \tau'\right)} - \log \frac{\exp\left(\langle f_\theta(\mathbf{Z}_i), \hat{g}_\theta(\phi_i) \rangle / \tau'\right)}{\sum_{j=1}^{n} \exp\left(\langle f_\theta(\mathbf{Z}_j), \hat{g}_\theta(\phi_i) \rangle / \tau'\right)}.$$

(5)

**Intra-parameter domain contrastive loss:** In order to fully exploit data potential within parameter space and better guide the learning of the representation and preserve similarities within the parameter space, we add an intra-parameter contrastive loss and find empirically that it helps accelerate the training process:

$$\ell_{\phi\phi}(\theta; \tau) := \frac{1}{n} \sum_{i=1}^{n} -\log \frac{\exp\left(\langle \hat{g}_\theta(\phi_i), \hat{g}_\theta(\tilde{\phi}_i) \rangle / \tau\right)}{\sum_{j=1}^{n} \exp\left(\langle \hat{g}_\theta(\phi_i), \hat{g}_\theta(\phi_j) \rangle / \tau\right)},$$

(6)

where $\tilde{\phi}$ is selected through data augmentation by perturbing $\phi$ with small amount of noise (see §B.1) or by finding nearest neighbors in the training set.

**Regression head:** Zimmermann et al. [2021] shows that under certain assumptions (see §A.2), the feature encoder $f_\theta$ implicitly learns to invert the data generating process $H$ up to an affine transformation, implying we can estimate $\phi$ via $h(f_\theta(\mathbf{Z}))$ for some linear function $h$. Inspired by their analysis, we add a regression head $h_\theta$ (a single fully-connected linear layer) in our framework, and we ensure the affine relationship between $h_\theta$ and $f_\theta$ by letting them share the same backbone up to the last layers before the output.

**Unified training procedure:** To this end, we formulate the final loss of our EMBED & EMULATE method as

$$\ell(\theta) = \lambda_{\mathbf{ZZ}} \ell_{\mathbf{ZZ}}(\theta; \tau) + \lambda_{\phi\phi} \ell_{\phi\phi}(\theta; \tau) + \lambda_{\mathbf{Z}\phi} \ell_{\mathbf{Z}\phi}(\theta; \tau') + \lambda_{\mathrm{MAPE}} \ell_{\mathrm{MAPE}}(\theta),$$

where $\lambda_{\mathbf{ZZ}}, \lambda_{\phi\phi}, \lambda_{\mathbf{Z}\phi}$ and $\lambda_{\mathrm{MAPE}}$ control relative loss weights. We show our basic setup in Fig. 1(a) and loss framework in Fig. 1(b).

# 5 Experiments

We conduct a numerical case study on the multiscale Lorenz-96 (L96) model [Lorenz, 1996], which is a common test model for climate models with both "fast" (high-frequency) and "slow" (low-frequency) components and other geophysical applications [Majda and Harlim, 2012, Law and Stuart, 2012, Law et al., 2016, Brajard et al., 2020]. It is a prototypical turbulent dynamical system "designed to mimic baroclinic turbulence in the midlatitude atmosphere" [Majda et al., 2010]. Its dynamics exhibit strong energy-conserving non-linearities, and for some settings of $\phi$, it can exhibit strong chaotic turbulence. Code is available at: https://github.com/roxie62/Embed-and-Emulate

The governing equations of the L96 system are defined as follows:

$$\frac{d\mathbf{X}_t^k}{dt} = -\mathbf{X}_t^{k-1}(\mathbf{X}_t^{k-2} - \mathbf{X}_t^{k+1}) - \mathbf{X}_t^k + F - hc\bar{\mathbf{Y}}_t^k,$$

$$\frac{1}{c}\frac{d\mathbf{Y}_t^{j,k}}{dt} = -b\mathbf{Y}_t^{j+1,k}(\mathbf{Y}_t^{j+2,k} - \mathbf{Y}_t^{j-1,k}) - \mathbf{Y}_t^{j,k} + \frac{h}{J}\mathbf{X}_t^k,$$

where $\mathbf{X}_t \in \mathbb{R}^K$ denotes the slow variable at the $t$-th time stamp and $\mathbf{Y}_t \in \mathbb{R}^{KJ}$ denotes the fast variable. We use $\mathbf{Z}_t := [\mathbf{X}_t, \mathbf{Y}_t] \in \mathbb{R}^{K(J+1)}$ to denote the system state at the $t$-th time stamp. We choose $K = 36$ and $J = 10$ throughout experiments in this section, as in Schneider et al. [2017].

We set our baseline following Schneider et al. [2017]. Within the EnKI framework, they use Runge-Kutta Dormand and Prince [1980] to compute the forward model $H$ and optimize (1). After analyzing the physics information of the L96 system, they define the moment function as: $m(\mathbf{Z}) := [\langle\mathbf{X}\rangle_T, \langle\bar{\mathbf{Y}}\rangle_T, \langle\mathbf{X}^2\rangle_T, \langle\mathbf{X}\bar{\mathbf{Y}}\rangle_T, \langle\bar{\mathbf{Y}}^2\rangle_T]$, where $\langle\cdot\rangle_T$ denotes an empirical average over $T$ time steps, $\bar{\mathbf{Y}}$ denotes an average across the $J$ $\mathbf{Y}$ channels, $\langle\bar{\mathbf{Y}}\rangle_T = (1/JT)\sum_{t=1}^T\sum_{j=1}^J\mathbf{Y}_t^{j,k} \in \mathbb{R}^K$, and other quantities are computed similarly (see details in §B.1). $m(\mathbf{Z}_t) \in \mathbb{R}^{5K}$. For the SBI methods, to suit the scenario we care about, i.e., estimating multiple $\phi_i$ for multiple different observations $\mathbf{Z}_i$ at test time, we compare with the non-adaptive counterpart of the algorithms referenced in these papers. For instance, we set the round number of SNL+ [Chen et al., 2020b] to be one. And use the fixed training dataset for SNL+ and NPE-C [Greenberg et al., 2019].

We conduct experiments and demonstrate the efficacy when measuring (1) the accuracy of averaged point estimates, (2) the accuracy of uncertainty quantification, (3) the empirical computational complexity. For all of our experiments (including SNL+ and NPE-C), we use ResNet-34 [He et al., 2016] as the backbone of the trajectory encoder. We apply average pooling at the last layer to generate a 512-d hidden vector. For contrastive learning, we project the 512-d hidden vector into a 128-d feature vector as the output of the trajectory encoder. For the regression head, we map the 512-d hidden vector to a 4-d parameter vector. It is important to note that activation functions are not used in the regression head to ensure an affine mapping between two projections. We parameterize the emulator network with a ResNet backbone, and to enhance the representation power of the network, we use five residual connection blocks (see §B.2). We find this increases the alignment between the two representation spaces. Within each skip block, there is a residual connection between the input layer and the output. The implementation details for training are shown in §B.1.

## 5.1 Higher quality estimates with lower sample cost

In this section, we evaluate our method with different training sizes in the perfect-model setting where no noise is injected in observations. These experiments demonstrate the usefulness of our method in realistic scenarios where forward models are prohibitively expensive and only limited quantities of training data are available.

We train our EMBED & EMULATE method and the supervised regression baselines when the training size equals 500 and 1,000. The training samples are sampled uniformly from $\phi_{\min} = [F_{\min}, h_{\min}, c_{\min}, b_{\min}] = [-5, 0, 0.1, 0]$ to $\phi_{\max} = [F_{\max}, h_{\max}, c_{\max}, b_{\max}] = [20, 5, 25, 25]$, and each simulation is of length 100, with $dt = 0.1$. We then sample 200 testing samples uniformly from a narrower range of $[-3, 0.5, 2, 2]$ to $[18, 4.5, 23, 23]$. As assumed in Schneider et al. [2017], each testing sample is of length 1,000, with $dt = 0.1$. The EnKI prior used for baselines (i.e. minimizing (1)) is a Gaussian distribution with means at the middle of the range of the testing instances, diagonal covariance matrix, and variances broad enough that all test samples are within $2\sigma$ of the mean, denoted $p_{\phi,\text{fixed}}$ (see §B.1). Within the context of the EMBED & EMULATE approach, we adopt an empirical Bayes approach: for each test instance, we compute $\hat{\phi}$ using the regression head and use

these values as the prior means when using EnKI to minimize (4); we denote this prior $p_{\phi,\mathrm{empB}}$. The impact of the empirical Bayes approach is explored in §5.3.

| $n$ | | $F\downarrow$ | $h\downarrow$ | $c\downarrow$ | $b\downarrow$ |
|---|---|---|---|---|---|
| 0 | EnKI *w/o* Learning | 15.48 (3.77) | 0.86 (0.20) | 40.45 (13.39) | 4.60 (0.60) |
| 500 | $h_\theta$ *w/* EMBED & EMULATE | 11.19 (3.65) | **3.18** (**1.60**) | **15.52** (6.24) | **8.57 (2.17)** |
| | EnKI *w/* EMBED & EMULATE | **10.94** (4.07) | 3.74 (2.26) | 16.09 (6.41) | 8.84 (2.93) |
| | Supervised Regression | 11.97 (**3.10**) | 4.07 (2.24) | 17.04 (**5.88**) | 9.07 (2.74) |
| | NPE-C | 15.51 (4.52) | 5.94 (2.59) | 23.54 (8.16) | 9.42 (3.70) |
| | SNL+ | 36.88(19.93) | 48.71 (30.69) | 57.45 (28.59) | 23.45 (17.62) |
| 1000 | $h_\theta$ *w/* EMBED & EMULATE | **6.30** (1.86) | **2.07 (1.31)** | **9.34** (3.71) | **5.51 (1.74)** |
| | EnKI *w/* EMBED & EMULATE | 6.59 (2.14) | 2.36 (1.54) | 9.38 (4.02) | 5.54 (2.35) |
| | Supervised Regression | 7.57 (**1.78**) | 3.08 (1.54) | 11.46 (**3.29**) | 6.64 (2.19) |
| | NPE-C | 11.29 (3.54) | 5.62 (2.29) | 16.32 (6.53) | 6.81 (2.38) |
| | SNL+ | 33.29(19.68) | 49.05 (27.75) | 48.17 (23.90) | 25.40 (16.03) |

Table 1: **Averaged MAPE (MdAPE, median absolute percentage error) for varying training size of different methods for 200 test samples.** We compare EMBED & EMULATE *w/* regression head ($h_\theta$) and EMBED & EMULATE plugged into EnKI to supervised regression, NPE-C [Greenberg et al., 2019], SNL+ [Chen et al., 2020b], and a classical numerical solver (Runge-Kutta) plugged into EnKI to minimize (1). The example illustrates that EMBED & EMULATE is able to achieve a lower error than both the EnKI approach based on a fixed moment vector objective and classical numerical solver [Schneider et al., 2017] and a straightforward supervised regression approach that is unable to produce uncertainty estimates.

| | Training data generation | Training time | 200 test runs | Total time (train + test) |
|---|---|---|---|---|
| EnKI *w/o* Learning | 0.0 | 0.0 | 8,000.0 | 8,000 (5.5 d) |
| $h_\theta$ *w/* EMBED & EMULATE | 21.0 | 72.0 | 1.0 | 94.0 (1.57 h) |
| EnKI *w/* EMBED & EMULATE | 21.0 | 72.0 | 2.0 | 95.0 (1.58 h) |
| Supervised Regression | 21.0 | 59.0 | 1.0 | 81.0 (1.35 h) |
| NPE-C | 21.0 | 72.0 | 2.0 | 95.0 (1.58 h) |
| SNL+ | 21.0 | 73.0 | 400.0 | 494.0 (8.23 h) |

Table 2: **Empirical computational time for different stages of different methods** ($n = 1,000$). Reported in minutes, total time for EMBED & EMULATE, the supervised regression, or NPE-C are 1.19% of EnKI with Runge-Kutta. All neural approaches are trained with 4 GPUs and tested with 1 GPU. Both training data generation and EnKI *w/* Learning are run with 32 CPU Cores.

We first evaluate the accuracy of point estimates using the mean and median absolute percentage error (MAPE and MdAPE), see §B.1. In Table 1 and Table 2, we see that EMBED & EMULATE evaluated with the regression head ($h_\theta$) and EnKI guarantees similar performance in terms of averaged accuracy. However, the regression head ($h_\theta$) is slightly better, which may be explained by our empirical observation that when estimates from $h_\theta$ with EMBED & EMULATE are far from the true parameter values, then using this estimate as the prior mean for EnKI can worsen the estimate. Despite this challenge, the EnKI has the advantage of providing uncertainty estimates, which are discussed below and reflected in Table 3.

Compared with other methods, it is clear that EMBED & EMULATE yields a significant improvement in accuracy, especially for the parameter affecting high-frequency dynamics (e.g., $c$). In particular, NPE-C [Greenberg et al., 2019] performs worse with a smaller training set size ; SNL+ [Chen et al., 2020b] using MCMC or rejection sampling suffers from increasing evaluation time as the number of test samples increases. Moreover, when compared to the classical method of running EnKI using a predefined moment function and expensive numerical solvers, EMBED & EMULATE yields better performance in terms of accuracy and computation time. Last, our EMBED & EMULATE framework also performs well relative to supervised regression for smaller training sets.

We then evaluate our results based on the continuous ranked probability score (CRPS) [Hersbach, 2000, Zamo and Naveau, 2018, Pappenberger et al., 2015]. CRPS is an important metric used in quantifying uncertainty (e.g., in weather forecasts). It measures the accuracy of estimated posterior distributions and is defined as

$$\text{CRPS}(C, \boldsymbol{\phi}_{*j}) = \int_{-\infty}^{\infty} (C(y_j) - U(y_j - \boldsymbol{\phi}_{*j}))^2 \mathrm{d}y_j,$$

where $\hat{\boldsymbol{\phi}}_{*j}$ represents the $j$-th component of the estimate parameter vector, $\boldsymbol{\phi}_{*j}$ represents the $j$-th component of the true parameter vector, $C$ is the cumulative density function of the ensemble estimates with $C(y) = P(\boldsymbol{\phi}_{*j} \leq y)$, and $U$ is the Heaviside step function. For a deterministic estimate from supervised regression, CRPS is equal to the mean absolute error (MAE). Table 3 shows that EMBED & EMULATE achieves high-accuracy estimates of uncertainty.

| $n$ | | $F \downarrow$ | $h \downarrow$ | $c \downarrow$ | $b \downarrow$ |
|---|---|---|---|---|---|
| 0 | EnKI *w/o* Learning | 0.910 | 0.019 | 2.443 | 0.393 |
| 500 | $h_\theta$ *w/* EMBED & EMULATE | 0.698 | 0.076 | 1.715 | 0.824 |
| | EnKI *w/* EMBED & EMULATE | **0.615** | **0.073** | **1.561** | **0.720** |
| | Supervised Regression | 0.707 | 0.104 | 1.785 | 0.917 |
| | NPE-C | 0.844 | 0.106 | 2.117 | 0.853 |
| | SNL+ | 1.399 | 0.616 | 2.426 | 1.822 |
| 1000 | $h_\theta$ *w/* EMBED & EMULATE | 0.412 | 0.049 | 0.992 | 0.453 |
| | EnKI *w/* EMBED & EMULATE | **0.360** | **0.042** | **0.829** | **0.394** |
| | Supervised Regression | 0.478 | 0.078 | 1.242 | 0.650 |
| | NPE-C | 0.593 | 0.096 | 1.389 | 0.564 |
| | SNL+ | 1.304 | 0.595 | 2.010 | 1.763 |

Table 3: **Continuous Ranked Probability Score (CRPS) evaluated on 200 test samples.** The errors of the uncertainty estimates are almost always lower for EMBED & EMULATE than for an EnKI method using a classical numerical solver. And compared to neural methods, EnKI with EMBED & EMULATE yields significant improvements over a supervised regression which cannot quantify uncertainty, defeats the simulation-based inference models NPE-C [Greenberg et al., 2019] and SNL+ [Chen et al., 2020b] which relies on sequential sampling. Moreover, it's clear that EMBED & EMULATE evaluated with EnKI provides more accurate uncertainty estimates than point-wise estimates from the regression head ($h_\theta$) trained with EMBED & EMULATE.

## 5.2 Visualizing uncertainty with noisy observations

In this subsection, we go beyond the perfect-model setting and evaluate our method in the realistic scenarios where observations are noisy and obtained through: $\mathbf{Z} = H(\boldsymbol{\phi}) + \boldsymbol{\eta}$, where $\boldsymbol{\eta} \sim \mathcal{N}(0, r\boldsymbol{\Gamma})$, where $\boldsymbol{\Gamma}$ is the temporal covariance of the trajectory $\mathbf{Z}$, and $r$ is a scaling value. Specifically, we set $\boldsymbol{\phi} = [10, 1, 10, 10]$ following Schneider et al. [2017] to ensure we are in the chaotic regime. We use model trained with $n = 4,000$ data samples and compare the results obtained in both the noiseless and noisy cases. For this visualization, both methods use the fixed prior from §5.1 based on the range of test values, so any differences observed are not caused by differences in the prior, but rather differences in the choice of objective and the method of computing the forward model.

As shown in Fig. 2, EMBED & EMULATE is more robust to noise than the baseline method with numerical solvers and predefined moments. Reconstructed posterior distributions learned with EMBED & EMULATE in Fig. 2 are more consistent with increasing noise levels, especially for the parameters $F$ and $b$.

## 5.3 Ablation study: role of the regression head

In this section, we empirically verify the utility of the regression head $h_\theta$ in our EMBED & EMULATE framework. We use $n = 4000$ training samples in two settings: first, we use the full EMBED & EMULATE model of §5.1; second, we discard the regression head (i.e. fix $h_\theta = 0$) while keeping intra- and inter-domain contrastive losses. For both trained models, we run the EnKI. We run the EnKI using components from the EMBED & EMULATE framework for two choices of prior: first, we use $p_{\boldsymbol{\phi}, \text{fixed}}$, and second, we use $p_{\boldsymbol{\phi}, \text{empB}}$. Note the empirical Bayes prior is only possible with the regression head. These priors are detailed in §5.1 and §B.1.

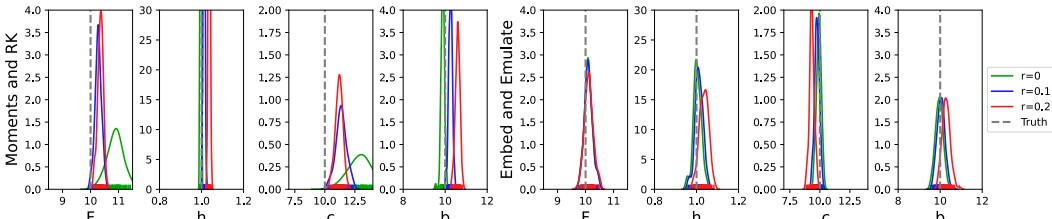

Figure 2: **Impact of observation noise.** Reconstructed posterior distributions, comparing a classical numerical solver (Runge-Kutta) plugged into EnKI to minimize (1) (left) with EMBED & EMULATE (right). Both variants of EnKI are run for 70 iterations with 100 particles. The green line shows the noise-free case, the blue shows the noisy case when $r = 0.1$, and the red shows the noisy case when $r = 0.2$. We see here that our proposed EMBED & EMULATE method (right column) produces posterior estimates that are consistent over a range of noise levels, while the baseline EnKI approach using a pre-defined embedding corresponding to a moments vector is much more sensitive to variations in $r$.

Table 4 shows that having the regression component of the loss complement the contrastive losses yields a substantial improvement in parameter estimation accuracy using EMBED & EMULATE. In other words, explicitly training the embedding function and emulator for the parameter estimation task yields an emulator that is far more accurate during parameter estimation than training a generic embedding and generic emulator. Furthermore, this table illustrates the efficacy of our empirical Bayes procedure – i.e., using our learned regressor to alter the prior used by EnKI. The medians are slightly improved while the means are strongly improved, suggesting that the empirical Bayes procedure is particularly helpful in the tails.

|  | $F \downarrow$ | $h \downarrow$ | $c \downarrow$ | $b \downarrow$ |
|---|---|---|---|---|
| (a) No regression head ($p_{\phi,\text{fixed}}$) | 16.75 (2.62) | 6.02 (1.63) | 22.16 (6.35) | 6.60 (1.86) |
| (b) EMBED & EMULATE ($p_{\phi,\text{fixed}}$) | 8.39 (1.12) | 1.77 (0.82) | 15.82 (1.61) | 3.85 (0.91) |
| (c) EMBED & EMULATE ($p_{\phi,\text{empB}}$) | **3.39 (1.03)** | **1.21 (0.76)** | **4.53 (1.52)** | **3.02 (0.90)** |

Table 4: **Ablation study of regression head:** Average MAPE (MdAPE, median absolute percentage error) over 200 test instances for estimating $\phi$ by minimizing (4) in three different settings: (a) $f_\theta$ and $\hat{g}_\theta$ correspond to a generic emulator trained without the regression loss $\ell_{\text{reg}}$ and the original prior $p_{\phi,\text{fixed}}$; (b) $f_\theta$ and $\hat{g}_\theta$ correspond to the emulator learned with our EMBED & EMULATE framework and the original prior $p_{\phi,\text{fixed}}$; and (c) $f_\theta$ and $\hat{g}_\theta$ correspond to the emulator learned with our EMBED & EMULATE framework and the empirical Bayes prior $p_{\phi,\text{empB}}$. This experiment shows that having the regression component of the loss complement the contrastive losses yields a substantial improvement in parameter estimation accuracy using EMBED & EMULATE.

## 6 Conclusions

The proposed EMBED & EMULATE framework trains an emulator of a complex simulation to facilitate parameter estimation. Unlike generic emulation methods, which can lead to poor parameter estimates and require expert domain knowledge to construct, our method (a) leverages a contrastive learning framework coupled with a regression head to jointly learn a low-dimensional embedding of simulator outputs that can be used to construct an objective function for parameter estimation and (b) yields an emulator that can be used within an optimization framework such as the EnKI to produce accurate parameter estimates in 1.19% of the computation time of an approach using classical numerical methods, even accounting for the time required to generate training samples. We explore our approach in the context of earth system modeling as described by Schneider et al. [2017] and Cleary et al. [2021] and hypothesize that these tools can facilitate improved climate forecasts that account for uncertainties and cover the full range of possible outcomes. The social impacts of improved climate forecasting are positive if acted upon. While learned emulators are gaining in popularity, we as a community still know little about which systems are more or less challenging to emulate or how to design task-specific emulators more generally. There are further opportunities for exploring emulators for parameter estimation using optimization frameworks beside the EnKI.

## Acknowledgments and Disclosure of Funding

We thank the anonymous reviewers and area chair for their helpful comments. We thank Owen Melia and Xiao Zhang for helpful discussion and proofreading our paper. RJ and RW are partially supported by DOE DE-AC02-06CH113575, DE-SC0022232, AFOSR FA9550-18-1-0166, NSF OAC-1934637, NSF DMS-2023109, and NSF DGE-2022023.

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
