# OpenReview forum: "Embed and Emulate: Learning to estimate parameters of dynamical systems with uncertainty quantification"
_NeurIPS.cc/2022/Conference — NeurIPS 2022 Accept_

### Official Review · Reviewer_25cv · 2022-07-10

**Rating:** 8
**Confidence:** 4
**Soundness:** 4 excellent
**Presentation:** 4 excellent
**Contribution:** 3 good

**Summary:**

The paper deals with the problem of Bayesian parameter estimation for high dimensional systems, where generating a large number of particles for ensemble Kalman inversion (EnKI) may be prohibitively expensive. The authors approach this by jointly learning a lower dimensional latent representation of the physical space as well as an emulator on this low dimensional latent space so that EnKI can be applied without significant computational cost (called _embed & emulate_). Ideas from contrasive learning are used to design the loss function for training, so that the trajectories that are close in the physical space gets pulled together in the latent space. The method is benchmarked on the multiscale Lorenz-96 model, where the _embed & emulate_ approach is compared against traditional EnKI and supervised regression in terms of runtime and several performance metrics.

**Questions:**

- Currently, the algorithm seems to only deal with state-space models where the observation operator is the identity map. However, in practice, more often than not the observations are not the identity map (e.g. observes only some components of $Z$, or maybe it is a nonlinear function of the state $Z$). Is it possible to handle more general observation operators to make it more useful in practice?
- How robust is this method to model mismatch? i.e., when the algorithm is trained on a physical model that differs from the underlying truth (e.g., when the algorithm is trained on a single-scale L96 model with unknown parameters $\phi = \\{h, c, F\\}$, but the underlying truth is the multiscale L96).
- Why are the same values for $\tau$ used in the losses $\ell_{ZZ}$ and $\ell_{\phi \phi}$ but a different value $\tau'$ is used for $\ell_{Z\phi}$?
- When computing $\hat{g}_\theta(\phi)$, do we not need to specify the initial condition $Z_0$? Since the trajectory $Z = H(\phi)$ depends on it.
- Not sure where the moments $m(Z)$ are used in the algorithm, even though it is mentioned.
- Can this inversion method be used in tandem with online filtering/smoothing (e.g. 4DVar, EnKF)?

**Limitations:**

- As with most deep learning applications, the approach is limited to the parameter distribution used in the training data; if the true parameter lie outside of this distribution, the method will likely struggle. Thus, one needs to cover a wide range of parameters, which may require a very large number of training data. This may be challenging for very high dimensional systems used in NWP.

- If the method cannot deal with general observation operators or not robust to model mismatch (see "Questions" above), this may be quite limiting for practical usage.

**Strengths And Weaknesses:**

Strengths:
- The paper presents an interesting and novel approach to learning useful low dimensional representations of high dimensional models. It is a nice idea to use this emulator for parameter estimation.
- The experiments demonstrate that the method captures good parameter estimates and uncertainties, outperforming EnKI with significantly less compute time. Moreover the approach is robust to noise.
- Experimental procedure/setup is clear and detailed.
- The paper is overall well-written

Weaknesses:
- Labels in the figures 2a and 3 can be larger. There is also an error in the caption of Figure 3:

> ''Embed & Emulate method _(top row)_ produces posterior estimates...''

There is no 'top row'.

- There are some passages in the text where more explanation is desired. More details below in "Questions".

---

> ### Author Response · Authors · 2022-08-01
> **Observation models and miscellaneous (P1)**
>
> We thank you for your comments and positive feedback on our paper. We address your specific concerns as follows.
>
> **Q1: ''Labels in the figures 2a and 3 can be larger. There is also an error in the caption of Figure 3.''**
>
> **A1**: Thank you for your suggestion. We will modify the arrangement of Figures 2(a) and 3, and correct the caption in the revision.
>
> **Q2: ''Currently, the algorithm seems to only deal with state-space models where the observation operator is the identity map. However, in practice, more often than not the observations are not the identity map (e.g. observes only some components of, or maybe it is a nonlinear function of the state). Is it possible to handle more general observation operators to make it more useful in practice?''**
>
> **A2**: Yes, we believe there are two potential ways to address the issue of partial and nonlinear observations.
> First, from a contrastive learning perspective, to handle missing data, we can extend our data augmentation by including random masking, and the goal of learning is to assign the representation of the masked data to the same cluster of unmasked ``anchor'' data so that the learned representation is invariant to such partial observations, as proposed in a recent work Assran et al. [2022].
> What's more, empirical studies show that a nonlinear transformation (often shown as corrupted data) of the true state can be well addressed by contrastive learning [Ravula et al., 2021]. Second, Chen et al. [2021], a recent work in Auto-differentiable Ensemble Kalman Filters (Auto-EnKF), shows empirical evidence of recovering underlying dynamics given partial observations when using a neural network as the surrogate model to approximate the dynamics evolution. Therefore, we are very likely to address the imperfect observation scenario and will consider this as future direction of interests.
>
> **Q3: ''How robust is this method to model mismatch? i.e., when the algorithm is trained on a physical model that differs from the underlying truth (e.g., when the algorithm is trained on a single-scale L96 model with unknown parameters with $\boldsymbol{\phi} = \\{h, c, b\\}$, but the underlying truth is the multiscale L96).''**
>
> **A3**: Our goal is to find the parameters of a physical model that best match observations. There is no assumption here that the observations were generated by the model or that there are ''true'' parameters. Our approach is akin to classical settings in which one uses derivative-free optimization to find the ''best'' model parameters that fit data; in this sense, *there is no model mismatch* in our setting. Learning to estimate model parameters does require data generated by that model, with corresponding parameters, and not some other model operating in a different parameter space; we cannot learn to estimate parameters we never see. That said, we speculate that it may be possible to train our system on a simpler model and then generalize it to a more complex model using ideas from transfer learning and curriculum learning.
>
> **Q4: ''Why are the same values for $\tau$ used in the losses $\ell_{\boldsymbol{\phi} \boldsymbol{\phi}}$ and $\ell_{\mathbf{Z} \mathbf{Z}}$ but a different value $\tau'$ is used for $\ell_{\boldsymbol{\phi} \mathbf{Z}}$?''**
>
> **A4**: Temperature values in contrastive learning influence (in terms of weights of gradients) of ''barely'' (''hard'') and ''easily'' (''easy'') distinguishable samples, as discussed in lines 630-638. We use $\tau$ to be the same for $\ell_{\boldsymbol{\phi} \boldsymbol{\phi}}$ and $\ell_{\mathbf{Z} \mathbf{Z}}$ as we assume the intra-domain pairwise relationship share similar structures. Under the perfect bijective relationship between the parameter and trajectory domain, the temperature value $\tau'$ influencing intra-domain losses should be set the same as $\tau$, and that is what we do for the initialization of $\tau'$ and $\tau$, and we keep them fixed for the first half of the training procedure. The reason we denote them differently is purely empirical, using the ''heating up'' strategy of Zhang et al. [2018] to improve the performance for the 2nd half training epochs. In this procedure, we find out changing $\tau$ and $\tau'$ together would destablize the performance, while keeping $\tau'$ fixed while only increasing $\tau'$ gradually will help improve the robustness of the performance.
>
> **Q5**: **''When computing $\hat{g}_\theta(\boldsymbol{\phi})$, do we not need to specify the initial condition $\mathbf{Z}_0$? Since the trajectory $\mathbf{Z} = H(\boldsymbol{\phi})$ depends on it.''**

---

> > ### Author Response · Authors · 2022-08-01
> > **Observation models and miscellaneous (P2)**
> >
> > **A5**: We do not need to specify the initial condition $\mathbf{Z}_0$ as our goal is to learn structural information of a time-series $\mathbf{Z}$, and we want the learned representation focuses more on the large window dynamics evolvement and to be invariant to the initial condition $\mathbf{Z}_0$. And to evaluate our argument, we independently sample the initial conditions of $\mathbf{Z}$ from a multi-normal distribution for each sample in the training and testing dataset.
> >
> > **Q6: ''Not sure where the moments $m(\mathbf{Z})$ are used in the algorithm, even though it is mentioned.''**
> >
> > **A6**: $m(\mathbf{Z})$ is used in the baseline of running EnKI with Runge-Kutta. A detailed explanation could be found from lines 562-567 in section A.1.
> >
> > **Q7: ''Can this inversion method be used in tandem with online filtering/smoothing (e.g. 4DVar, EnKF)?''**
> >
> > **A7**: We believe that this may be possible, potentially leveraging ideas from Chen et al. [2021]. This is beyond the scope of the current submission, but an active area of interest. Such an approach may also make it possible for our emulation ideas to be applied in settings in which we have incomplete data, as referred to in **Q2**.”
> > Extensions to methods like 4D-VAR could be quite interesting and are an important area of ongoing work.
> >
> > **Q8: ''As with most deep learning applications, the approach is limited to the parameter distribution used in the training data; if the true parameter lie outside of this distribution, the method will likely struggle. Thus, one needs to cover a wide range of parameters, which may require a very large number of training data. This may be challenging for very high dimensional systems used in NWP.''**
> >
> > **A8**: Yes, out-of-domain sampling is not our focus in this paper.
> > But when compared to the baseline model, a huge advantage of our method is that we could amortize the sampling and training cost across the multiple different (testing) data points, and is able to repeatedly carry out inference for new arriving data without incurring any sampling or training cost further.
> > As pointed out in our manuscript, the range of values we use to generate our training data is *significantly* larger than that of previous work in this space.
> > Furthermore, climate science experts (e.g., at the National Center for Atmospheric Research) tell us that parameter estimation is often conducted on low-dimensional subsets of the full parameter space (in part due to the complexity of running their simulators repeatedly within classical derivative free optimization frameworks), making the generation of sufficient training data tractable. In fact, given the computational advantages of our proposed approach, it's possible that our method may be able to operate in higher-dimensional parameter spaces than classical methods.
> >
> > **Q9: ''If the method cannot deal with general observation operators or not robust to model mismatch (see "Questions" above), this may be quite limiting for practical usage.''**
> >
> > **A9**: As discussed above, we believe our model could be adapted to general observation operators and model mismatch is not a concern for our setting. Furthermore, the parameter estimation problem as we have formulated it is widely used in a diverse collection of settings in the physical sciences.
> >
> > **References**
> >
> > Mahmoud Assran, Mathilde Caron, Ishan Misra, Piotr Bojanowski, Florian Bordes, Pascal Vincent,Armand Joulin, Michael Rabbat, and Nicolas Ballas. Masked siamese networks for label-efficient learning. *arXiv preprint arXiv:2204.07141*, 2022.
> >
> > Sriram Ravula, Georgios Smyrnis, Matt Jordan, and Alexandros G Dimakis. Inverse problems381
> > leveraging pre-trained contrastive representations. *Advances in Neural Information Processing Systems*, 34, 2021
> >
> > Yuming Chen, Daniel Sanz-Alonso, and Rebecca Willett. Auto-differentiable ensemble kalman filters. *arXiv preprint arXiv:2107.07687*, 2021.
> >
> > Xu Zhang, Felix Xinnan Yu, Svebor Karaman, Wei Zhang, and Shih-Fu Chang. Heated-up softmax
> > embedding. *arXiv preprint arXiv:1809.04157*, 2018.

---

> > > ### Comment · Reviewer_25cv · 2022-08-09
> > > **Thank you for clarifying**
> > >
> > > I thank the authors for taking the time to respond to all the reviews in great details. I believe all of my points have been addressed. In particular, the techniques pointed out for addressing partial/nonlinear observation models sound promising and would be interested to see more in the future. The additional experiments conducted by the authors to address the other reviewers' concerns are also sound and I appreciate their effort and time in doing this.
> > >
> > > Overall, I personally find this work very exciting and opens up new possibility for learning from very high dimensional data. In light of the author response, I have a better appreciation of the work and raise the score from 7 to an 8.

---

### Official Review · Reviewer_fsU3 · 2022-07-11

**Rating:** 7
**Confidence:** 3
**Soundness:** 3 good
**Presentation:** 3 good
**Contribution:** 2 fair

**Summary:**

The paper proposes a parameter estimation approach for (non-linear) dynamical systems with multidimensional outputs. The key novelty involves jointly learning a low-dimensional embedding of the observations (Z) and an emulator mapping from parameters to a representation in the latent space while ensuring that the two representations are aligned and using contrastive learning. The learned representations can now be used in a classical parameter estimation setup such as EnKI. The method is evaluated on the L96 system, and the results are compared to a standard parameters estimation technique (EnKI with numerical solvers). The proposed method compares favorably in terms of both MAPE and CRPS to the chosen baselines on the L96 problem (and the additional problem in the appendix). Further experiments are included to demonstrate the uncertainty quantification properties and importance of specific elements of the method.

**Questions:**

Minor questions/comments/suggestions:

- Choice of EnKI: Early on the paper makes a choice to use EnKI, and while this appears sensible there are several other derivative free optimization methods. I’d suggest strengthening the justification for the EnKI even further and perhaps clarify (/speculate) how sensitive the proposed E&E approach is to using EnKI.

- Resnet-32 structure: I would suggest clarifying why a convolutional resnet is a suitable backbone for the particular problem.

- l197: For completeness, please provide the reference for InfoNCE and explain the modifications.

- l199: $p$ appears to be undefined (?)

- l201: Please clarify which metric is actually used (i.e. $\delta$)? Also, how sensitive is the method to the choice of $\delta$ ?

- l211 “…due to the dimensions of the dynamics to be emulated (i.e. dimensions of Zs)”. I appreciate that the dimension of Z is challenging but does it make sense to talk about it as the “dimensions of the dynamics”…?

- Optimization problem (l237-238): could the authors comment on the difficult of optimizing this objective (in terms of robustness, sensitivity to hyperparameters of the optimization algorithms etc.) and perhaps describe/show some optimization traces.

- l260: “latent” seems more appropriate to me than “hidden” to me.

- Eq 3: \langle \rangle are used for an inner product (I presume); later used as empirical average l253; perhaps clarify c.f. eq. 3 to avoid confusion.

- Discussion: The current method requires quite a lot of training data which is expensive to generate; do the authors see the current method being used in a sequential experimental design (aka active learning) setup?


**Limitations:**

I think the paper could do a slightly better job explaining the complexity and generalizability of the approach (see comments elsewhere).

**Strengths And Weaknesses:**


General comments/suggestion/questions:

- A mostly clear and well-written paper on a timely problem. The resulting method is rather complex with many parts, but I think the paper does a relatively good job of guiding the reader through the various elements.

- A seemingly clear limitation is the inclusions of only one empirical study, namely based on the L96 Lorentz system; however, the experiment appears well-executed and well-documented, so I do not see any mainly issues with only a single case study given the nature of the problem (assuming the below question is addressed). I note the additional experiment in the appendix also favors the proposed method over the considered alternatives, and the authors may want to refer to the additional experiment in the main paper.

- Complexity, practicality, and generality is my main concern with the proposed method. The rationale used to motivate the need for the E&E method focuses (initially) on the need to make choices about moments in the cost functions; however, the proposed method is very complex by any standard. Consequently, I would argue that there are even more design choices involved in the E&E method ranging from the backbone, dimensionality of latent embedding, metrics in the latent space, hyperparameters, etc. Could the authors clarify how general the specific method (really) is and to which degree it needs to be tailored to each specific problem (domain)?

---

> ### Author Response · Authors · 2022-08-01
> **Complexity, practicality, and generality (P1)**
>
> We thank you for your comments and positive feedback on our paper. We address your specific concerns as follows.
>
> **Q1: ''A seemingly clear limitation is the inclusions of only one empirical study, ... I note the additional experiment in the appendix also favors the proposed method over the considered alternatives, and the authors may want to refer to the additional experiment in the main paper.''**
>
> **A1**: We thank you for your suggestion. We will add reference to the additional experiment in the revision of the main paper.
>
> **Q2: ''Complexity, practicality, and generality is my main concern with the proposed method. The rationale used to motivate the need for the E\&E method focuses (initially) on the need to make choices about moments in the cost functions; however, the proposed method is very complex by any standard. Consequently, I would argue that there are even more design choices involved in the E\&E method ranging from the backbone, dimensionality of latent embedding, metrics in the latent space, hyperparameters, etc. Could the authors clarify how general the specific method (really) is and to which degree it needs to be tailored to each specific problem (domain)?''**
>
> **A2**: We believe Embed \& Emulate is practically reliable and easily adapts to different problems. The best evidence of this is that our system works well not only for the L96 experiments in the main paper, but also for the KS experiments in the supplement with minimal adjustments. In contrast, it is not at all clear how to set an objective function for the KS setting without expert knowledge.
> We don't have a simple parametric form for objective functions, so choosing a good objective function is generally more complex than hyperparameter tuning. Our system, while complex, was tuned over a small number of values for a small number of hyperparameters, all of which could be set reasonably well using reasonable heuristics. For example, we developed our framework using the L96 system; we then applied it to the KS system, and the only change required was altering the embedding dimension.
>
> More specifically, our architecture uses a ResNet backbone -- a well-known powerful network suited to many downstream tasks in computer vision, especially for contrastive representation learning. Its efficacy in varied settings suggest it is a good, general-purpose backbone that would not need to be redesigned for a new parameter estimation problem due to its representation power for complex and dynamic data. And we can easily adjust the size of the Resnet to correspond to the dimensionality of the data.
> Second, the dimensionality of the latent embedding is generally chosen in an empirical way using grid search, a simple heuristic is to set it to be $8, 12$, or  $16$ times the number of dimensionality of $\boldsymbol{\phi}$. The rationale behind this rule is that, under a few assumptions (see in Section A.2), the embedding of the learned contrastive representation would be an affine transformation $A$ (with full column rank) of the inverse of the data generating process $H^{-1}(\mathbf{Z})$ [Zimmermann et al., 2021].
> Therefore, setting the embedding dimension a few times larger than the parameter dimension will be enough to approximate the inverse mapping $A^{-1}$. Similarly, the dimensionality of latent space is always selected empirically between $\{128, 256, 512\}$ and is normally set to be a bit larger than the input and output dimension. Third, only two important hyperparameters arisen with contrastive representation learning are temperature value $\tau$, number of negative samples (memory bank size). As discussed in Section B.1 from line 630 to 638, temperature values balance the influence between barely (hard) and easily (easy) distinguishable. We  adopt the heat-up strategy of Zhang et al. [2018], start with a low $\tau'$, monitor the decrease of the training loss, and linearly increase $\tau'$ toward the end of training to add impacts of ``easy'' samples. Lastly, we should set the number of negative samples to be as large as possible to ensure better empirical performance [Wang and Isola, 2020].

---

> > ### Author Response · Authors · 2022-08-01
> > **Complexity, practicality, and generality (P2)**
> >
> > **Q3: ''Choice of EnKI: Early on the paper makes a choice to use EnKI, and while this appears sensible there are several other derivative free optimization methods. I’d suggest strengthening the justification for the EnKI even further and perhaps clarify (/speculate) how sensitive the proposed E\&E approach is to using EnKI.''**
> >
> > **A3**: Great question! Particle-based methods like the EnKI are widely used for large-scale simulation parameter estimation, including in climate problems that help motivate our manuscript. Under the assumption of Gaussian noise $\boldsymbol \eta$ and linearity assumption of the forward operator, the EnKI is able to provide systematic uncertainty quantification in that the ensemble distribution constructed with EnKI is proved to be ''equivalent'' to the target posterior distribution [Ding et al., 2020]. Although linearity of forward operators is generally restrictive, we show in Section A.2 that the representation $f_\theta$ learned in a contrastive framework is an affine transformation of $H^{-1} (\mathbf{Z})$ where $H$ is the physical simulator; this fact supports us relying on the EnKI as a derivative free optimization method capable of providing uncertainty quantification.
> >
> > We speculate that particle filtering based approaches, while quite flexible, would not be as effective because of their well-documented issues in high-dimensional settings. Extensions to methods like 4D-VAR could be quite interesting and are an important area of ongoing work.
> >
> > **Q4: ''Resnet-32 structure: I would suggest clarifying why a convolutional resnet is a suitable backbone for the particular problem.''**
> >
> > **A4**: Detailed discussion is in **A2** and will be added in the revision. In addition, we think the \textit{convolutional} layer, a core component of ResNet, is a generic and natural embedding network for many physics simulators, in part because of the smoothness of the data over time and correlations between the channels.
> > Also, it should be displayed as ResNet34 in line 259.
> >
> > **Q5: ''l197: For completeness, please provide the reference for InfoNCE and explain the modifications.''**
> >
> > **A5**: Info Noise Contrastive Estimation (InfoNCE) [Oord et al., 2018] uses categorical cross-entropy loss to identify a ''positive'' sample $\tilde{\mathbf{Z}}_i$ among a set of ''negative'' samples $\\{\mathbf{Z}_j, j=1,\dots n\\}$, and they aim to learn the score function $s_\theta (\mathbf{Z}_i, \mathbf{Z}_j)$ measuring the similarity between any pair of data, while we define the similarity score as $\langle f_\theta(\mathbf{Z}_i), f_\theta(\mathbf{Z}_j) \rangle $ where $f_\theta(\mathbf{Z})$ is normalized and the goal is to learn the representation $f_\theta$. This information will be added to the revision.
> >
> > **Q6: ''l199: $p$ appears to be undefined (?)''**
> >
> > **A6**: $p$ at line 199 is the dimension of the learned embedding function.
> >
> > **Q7: ''l201: Please clarify which metric is $\delta$ actually used (i.e. )? Also, how sensitive is the method to the choice of $\delta$?''**
> >
> > **A7**: $\delta(\boldsymbol{\phi}_i, \boldsymbol{\phi}_j) := \frac{1}{2} \{ \rm{APE} (\boldsymbol{\phi}_i; \boldsymbol{\phi}_j) + \rm{APE} (\boldsymbol{\phi}_j; \boldsymbol{\phi}_i)\}$ where $\rm{APE}$ is short for absolute percentage error (in Section B.1 at line 616, lines 689-692). We use $\delta$ as an empirical metric of distance between any data points to selecting the ``positive'' sample (one is of the minimum $\delta$ value). As long as $\delta$ reflects the distance in parameter domain, our method will work well.
> >
> > **Q8: ''l211 '…due to the dimensions of the dynamics to be emulated (i.e. dimensions of Zs)'. I appreciate that the dimension of Z is challenging but does it make sense to talk about it as the 'dimensions of the dynamics'…?''**
> >
> > **A8**: Thanks for your suggestion, we will modify it as ``the dimensionality of observed (simulated) dynamical variables''.
> >
> > **Q9: ''Optimization problem (l237-238): could the authors comment on the difficulty of optimizing this objective (in terms of robustness, sensitivity to hyperparameters of the optimization algorithms etc.) and perhaps describe/show some optimization traces.''**
> >
> > **A9**: Our method, with objective function at line 238, is robust to hyperparameter tuning. In Table 3 below, we show the empirical results of making changes to our default setting when $n=1,000$; this table shows low errors across a range of hyperparameter values. The optimization trace can be a little counterintuitive because, when we heat up the parameter $\tau$ for the contrastive loss, the loss function is changing with epoch. However, we keep $\tau$ fixed for the first 500 epochs, and display the optimization trace for that phase of training in Table 4.

---

> > > ### Author Response · Authors · 2022-08-01
> > > **Complexity, practicality, and generality (P3)**
> > >
> > > |   | $F\downarrow$ | $h\downarrow$ | $c\downarrow$ | $b\downarrow$ |
> > > |------------|------------|------------|------------|------------|
> > > | Default | **5.23(2.27)** | 2.77(2.04) | 7.96(3.52) | 4.02(2.15) |
> > > |Fixed $\tau = 0.1$ | 5.60(2.62) |  3.15(2.11) | 7.40(4.30) | 4.03(2.32) |
> > > |Memory bank size $= 1,000$ | 5.99(2.37) | **2.46(1.77)** | 7.69(3.42) | 4.16(2.36) |
> > > |Initial learning rate = 0.001 | 5.71(2.77) | 2.95(2.02) | **6.84**(3.89) | 4.83(2.67) |
> > > |Embedding dimension $p = 48$ | 6.19(2.33) | 2.76(1.78) | 8.04(**3.04**) | **3.71(2.04)**|
> > >
> > > Table 3: **Hyperparameters sensitivity analysis: Average MAPE (MdAPE, median absolute percentage error) evaluated on 200 test samples when $n = 1,000$.** The default setting is: heat up $\tau$ from $0.1$ to $0.5$, set memory bank of size $2,000$, initialize the learning rate at $0.01$ and set the embedding dimension $p = 64$. At each row, we make one change to the default setting. This experiment show performance of EnKI with Embed \& Emulate is robust to hyperparameter tunning.
> > >
> > >
> > > |   | 0 | 50 | 100 | 150 | 200 | 250 | 300 | 350 | 400 | 450 | 500 |
> > > |---|---|---|---|---|---|---|---|---|---|---|---|
> > > |$\ell$ | 27.41| 22.28 | 18.92 | 17.65 | 15.86 | 15.01 | 14.65 | 13.73 | 14.08 | 12.63 | 12.28 |
> > >
> > > Table 4: **Optimization trace for first 500 epochs when $n = 1,000$ with $\tau = \tau' = 0.1$.**
> > >
> > > **Q10: ``l260: “latent” seems more appropriate to me than “hidden” to me.''**
> > >
> > > **A10**: Thanks for your suggestion. We will modify it in the revision.
> > >
> > > **Q11: ``Eq 3: $\langle \cdot \rangle$ are used for an inner product (I presume); later used as empirical average l253; perhaps clarify c.f. eq. 3 to avoid confusion.''**
> > >
> > > **A11**: We should use $\langle \cdot \rangle_T$ at line 253-255 to denote the empirical average.
> > >
> > > **Q12: ``Discussion: The current method requires quite a lot of training data which is expensive to generate; do the authors see the current method being used in a sequential experimental design (aka active learning) setup?''**
> > >
> > > **A12**: Active learning approaches such as those described in Settles [2009], Michael [2006] actively generate training data for a *single* test instance. Thus, over hundreds or thousands of test instances, a huge quantity of training data must be generated. In contrast, our approach yields a global emulator which can be used for many test instances, so the total amount of training data generated (across all test instances) is far smaller with our approach than with active learning approaches.
> > >
> > > More generally, active learning approaches are quite interesting. How effective active learning approaches may be in our setting, where we wish to learn a parameter estimator that can be used for many test samples, is an open and exciting question.
> > >
> > > **References**
> > >
> > > Roland S Zimmermann, Yash Sharma, Steffen Schneider, Matthias Bethge, and Wieland Brendel. Contrastive learning inverts the data generating process. *In International Conference on Machine Learning*, pages 12979–12990. PMLR, 2021.
> > >
> > > Xu Zhang, Felix Xinnan Yu, Svebor Karaman, Wei Zhang, and Shih-Fu Chang. Heated-up softmax embedding. *arXiv preprint arXiv:1809.04157*, 2018.
> > >
> > > Aaron van den Oord, Yazhe Li, and Oriol Vinyals. Representation learning with contrastive predictive coding. *arXiv preprint arXiv:1807.03748*, 2018.
> > >
> > > Zhiyan Ding, Qin Li, and Jianfeng Lu. Ensemble kalman inversion for nonlinear problems: weights, consistency, and variance bounds. *arXiv preprint arXiv:2003.02316*, 2020.
> > >
> > > Tongzhou Wang and Phillip Isola. Understanding contrastive representation learning through alignment and uniformity on the hypersphere. *In International Conference on Machine Learning*, pages 9929–9939. PMLR, 2020.
> > >
> > > Shuang Ma, Zhaoyang Zeng, Daniel McDuff, and Yale Song. Active contrastive learning of audio-visual video representations. *arXiv preprint arXiv:2009.09805*, 2020.

---

### Official Review · Reviewer_QCnJ · 2022-07-11

**Rating:** 6
**Confidence:** 3
**Soundness:** 3 good
**Presentation:** 2 fair
**Contribution:** 2 fair

**Summary:**

This study proposes a method that aims at efficiently solving the inverse problem of stochastic simulators. The new methodology uses neural networks to learn both an embedding of the simulator output and an emulator of the simulator -- i.e., a mapping from the parameters to the lower dimensional projection of the simulator output. The study provides empirical results on the multiscale Lorenz-96 (L96) model and shows that the new method leads to parameters that are closer to the ground-truth parameters, and with lower computational cost, compared to previous approaches.


**Questions:**

This study will substantially benefit from a thorough discussion and empirical comparison with previous methods for likelihood-free inference.

**Limitations:**

The authors have adequately addressed the limitations of their work.


**Strengths And Weaknesses:**

### Originality

The problem of efficiently and reliably inferring the parameters of stochastic simulators given empirical data is an important problem across fields of science. And, as far as I can tell, the solution provided by the study is technically novel. However, I have a few concerns:

-there is a large body of work relevant to this study that is currently not referenced, discussed and empirically compared. Indeed, methods that go under the name of Approximate Bayesian Computation, likelihood-free inference, or/and simulation-based inference would be very relevant to discuss and empirically compare. All these methods are precisely concerned with performing Bayesian inference of stochastic simulators (for a review, see https://www.pnas.org/doi/10.1073/pnas.1912789117). In particular, and most directly relevant to this study, several of these methods use neural density estimators and aim at estimating the likelihood function (i.e., an emulator - e.g., Papamakarios et al. 2019 http://proceedings.mlr.press/v89/papamakarios19a.html; Lueckmann et al. 2019 https://proceedings.mlr.press/v96/lueckmann19a.html), extracting informative summary statistics from the data for posterior inference (e.g. https://openreview.net/forum?id=SRDuJssQud), or directly estimating the posterior distribution (https://proceedings.neurips.cc/paper/2016/hash/6aca97005c68f1206823815f66102863-Abstract.html, https://proceedings.mlr.press/v97/greenberg19a.html). I believe it would be crucial to discuss and empirical compare the approach developed to at least a few of those previous approaches. I should note that there is code available to perform such comparisons (e.g. https://sbi-benchmark.github.io/);

-related to the point above, I believe the baseline approach provided, which the authors use to validate their approach, could have been better chosen. In particular, regarding the baseline approach, the authors write "However, it does not offer uncertainty estimates". A potentially stronger alternative would have been Neural Posterior Estimation (NPE), for instance with normalising flows (https://proceedings.mlr.press/v97/greenberg19a.html). NPE would likely require more simulations, but would provide a full posterior distribution and have the added advantage of allowing to perform amortised inference after network training.


Therefore, as it stands, the contribution of this study is not sufficiently put in the context of past work on this topic, and I would strongly recommend the authors to compare their method with some of these past methods.


### Quality

The paper is technically sound, and claims are backed by empirical evidence.


### Clarity

The manuscript is for the most part clearly written, providing enough information to understand the technical contribution and empirical results. A few small comments and typos:

-page 1, line 28, "$\eta$ is observation noise". Unless the expression in line 27 is incorrectly written, $\eta$ would be more appropriately described as "dynamics noise" rather than "observation noise", since it impacts the dynamics update equation;

-page 3, lines 113-114, "Cleary et al. [2021] is designed for inferring..." This formulation requires a small rewriting for clarity;

-page 3, line 139, "when the model H is bijective". As it is written, it suggests that many models are bijective. However, this is rather an exception, so I would suggest the authors to rephrase this;

-page 4, line 174, "the numerical algorithm used to simulate H will generally produce some errors". It would be beneficial if the authors provided a quantification (an appendix figure?) of such errors;

-page 5, line 197, "variant of InfoNCE". For clarity, it would be good to introduce the abbreviation;

-page 6, caption Figure 1, "intra-trajecotry";

-page 6, caption Figure 1, a few typos in "diagonals are corresponds to dot product".


### Significance

The technical development and empirical results will be of interest to the ML community.

---

> ### Author Response · Authors · 2022-08-01
> **Additional literature review and comparisons (P1)**
>
> **Q1: Questions regarding validating our approach with past works in approximate Bayesian computation (ABC), likelihood-free inference, or/and simulation-based inference (SBI).**
>
> **Q1-1: ''there is a large body of work relevant to this study ... All these methods are precisely concerned with performing Bayesian inference of stochastic simulators (for a review, see [Cranmer et al., 2020]).''**
>
> **A1-1**: Thank you for the thorough, explicit, and helpful list of papers. We will certainly improve our literature review during revisions accordingly. We did not originally cover this branch of the literature [Papamakarios et al., 2019, Cranmer et al., 2020, Lueckmann et al., 2021, Chen et al., 2020, Alsing et al., 2019] because, while both it and our manuscript are concerned with parameter estimation associated with physics-based simulators, they are designed with very different goals in mind. Specifically, our goal in learning an emulator is to replace costly simulators used in parameter inference for *multiple* different test observations $\mathbf{Z}_i$ with different underlying parameters $\boldsymbol{\phi}_i$, and we demonstrate that Embed \& Emulate yields significant computational advantages over methods that require using the simulator at test time. At test time, we perform Bayesian inference iteratively (using our learned emulator) to closely approximate the true posterior without re-simulating the data and re-training the model.
>
> In contrast, advanced SBI methods reviewed in Lueckmann et al. [2021] focus on adaptive generation of new training data (which
> requires running the simulator many times) *for each new test sample*. This approach may not compare favorably in terms of computational complexity with classical methods that do not depend on generating training data or training an emulator.
> (An exception to this is neural posterior estimation (NPE [Greenberg et al., 2019]); see A1-2 for details and empirical comparisons.)
> Furthermore, we specifically target parameter inference in high-dimensional data regimes with complex and chaotic dynamical behavior, whereas many works in SBI suffer from low accuracy in this regime. (Exceptions like NPE [Greenberg et al., 2019] and SNL+ [Chen et al., 2020] will be discussed in A1-2.)
>
> **Q1-2: Questions regarding relevant work ''several of these methods use neural density estimators and aim at estimating the likelihood function (i.e., an emulator-Papamakarios et al. [2019], Lueckmann et al. [2019]), extracting informative summary statistics from the data for posterior inference [Chen et al., 2020], or directly estimating the posterior distribution [Papamakariosand Murray, 2016, Greenberg et al., 2019].'' And baseline: ''related to the point above... NPE would likely require more simulations, but would provide a full posterior distribution and have the added advantage of allowing to perform amortized inference after network training.''**
>
> **A1-2**: To suit the scenario we care about, i.e., estimating multiple $\boldsymbol{\phi}_i$ for multiple different observations $\mathbf{Z}_i$ at test time (as discussed in A1-1), and for a fair comparison, we compare with the non-adaptive counterpart of the algorithms referenced in these papers (i.e., we have a fixed set of training samples used to train a system used for all test samples). For instance, we set the round number of SNL+ to be 1.
>
> Next, while our problems are in a high-dimensional regime (e.g., $\rm{dim}(\mathbf{Z}) = 396,000$ for Lorenz 96), sequential neural likelihood estimation (SNL) in Papamakarios et al. [2019] does not scale to the high-dimensional setting, which is recognized as a limitation and future direction by the authors. The later work in Chen et al. [2020] SNL+ addresses this limitation by learning sufficient statistics (embeddings) of the data based on the infomax principle. We show detailed empirical comparisons in Tables 1 and 2 for SNL+.
>
> Closely related to Papamakarios et al. [2019], Lueckmann et al. [2019] tries to ``emulate'' the simulator likelihood and advocates the use of active learning to choose the simulations in the next round based on uncertainty in the posterior. We are unable to compare with this method in this rebuttal as Lueckmann et al. [2019] does not appear to have available code and also lacks evidence that it can scale to high-dimensional data settings (their high-dimensional setting is of dimension $O(10^3)$ while ours is $O(10^6)$).
>
> Finally, sequential neural posterior estimation (SNPE-A in Papamakarios and Murray [2016], SNPE-C in [Chen et al., 2020, Greenberg et al., 2019]) trains a neural network to approximate the posterior distribution directly. SNPE-C gains an advantage against SNPE-A in that it has fewer restrictions regarding the form of prior and posterior by leveraging neural conditional density estimation [Papamakarios et al., 2017]. The detailed empirical comparisons regarding non-sequential SNPE-C (i.e., NPE-C) are in Tables 1 and 2.

---

> > ### Author Response · Authors · 2022-08-01
> > **Additional literature review and comparisons (P2)**
> >
> > **A1-2**: Setting the number of training samples  to 1000 (and hence equalizing the amount of simulation computation time), Embed \& Emulate consistently outperforms NPE-C and SNL+ in our L96 experiments. While NPE-C has the ability to implicitly learn ''good'' features from data in the process of mapping $\mathbf{Z}$ to the posterior, from Table 1 and Table 2, we observe that the method is particularly poor relative to Embed \& Emulate for estimating parameters controlling high-frequency dynamics. While SNL+, like Embed \& Emulate, learns an embedding using the infomax principle, SNL+ only cares about inter-domain relations and ignores the benefits of learning information about the intra-domain structure (i.e. the relationships between parameter vectors and observation vectors). In contrast, Embed \& Emulate using $\ell_{\boldsymbol{\phi} \boldsymbol{\phi}}$ and $\ell_{\mathbf{Z} \mathbf{Z}}$ excels at learning structural information and is well suited to learning high-frequency parameters.
> >
> > |             | F $\downarrow$ | h $\downarrow$ | c $\downarrow$ | b $\downarrow$ |
> > |------------ | -------------|-------------|-------------|-------------|
> > |NPE-C | 1.186| 0.062 | 2.933 | 0.523 |
> > | SNL+ | 0.647| 0.352 | 1.562 | 2.189 |
> > | EnKI with Embed \& Emulate | **0.345** | **0.058** | **0.721** | **0.331** |
> >
> > Table 1: **Continuous Ranked Probability Score (CRPS) evaluated on 200 test samples when $n=1,000$.** The errors of the uncertainty estimates are always lower for Embed \& Emulate than for NPE-C [Greenberg et al., 2019] and SNL+ [Chen et al., 2020].
> >
> > |             | F $\downarrow$ | h $\downarrow$ | c $\downarrow$ | b $\downarrow$ |
> > |------------ | -------------|-------------|-------------|-------------|
> > |NPE-C | 23.48(11.41)| 3.21(2.27)| 63.15(28.48) | 6.37 (3.79) |
> > | SNL+ | 9.68(8.73)| 29.20(18.53) | 20.67(16.26) | 36.87(26.40) |
> > | EnKI with Embed \& Emulate | **5.23(2.27)** | **2.77(2.04)** | **7.96(3.52)** | **4.02(2.15)** |
> >
> > Table 2: **Average MAPE (MdAPE, median absolute percentage error) evaluated on 200 test samples when $n=1,000$**. This experiment shows that Embed \& Emulate yields a substantial improvement in parameter estimation accuracy compared to NPE-C [Greenberg et al., 2019] and SNL+ [Chen et al., 2020].
> >
> > We will add the above discussion in the revision.
> >
> > **Implementation details**: We emphasize that the comparisons in Tables 1 and 2 were not trivial, as NPE-C and SNL+ could not be used ``out of the box'' in our setting. We needed to leverage several insights from our manuscript to achieve the performances reported in the tables. Specifically:
> >
> > We test both masked autoregressive flow (MAF) [Papamakarios et al., 2017] and Neural Spline Flow (NSF) [Durkan et al., 2019] as density estimation for NPE-C [Greenberg et al., 2019]. We find NPE-C with MAF outperforms NPE-C with NSF for the current problem setting and report the best result of NPE-C with MAF in Tables 1 and 2. We use masked autoregressive flow (MAF) for SNL+ [Chen et al., 2020].
> >
> > (1) We make the following changes to the codebase of ${\rm NPE}$ and ${\rm SL^+}$:
> >
> > For a fair comparison, we replace the default Convolutional Neural Network (CNN) with the same ResNet architecture in Embed \& Emulate and supervised regression, and use it as the backbone of the embedding network.
> >
> > To facilitate large-scale data (in terms of batch size and dimensionality of the data) training, we add a distributed script into the codebase to provide synchronous distributed training on multiple GPUs.
> >
> > We replace the default Adam optimizer with AdamW and add the cosine decay scheduler to gradually decay the learning rate.
> >
> > (2) The rule of selecting hyperparameters is using grid search (same as Embed \& Emulate and supervised regression):
> >
> > We choose the dimension of the embedding using grid search in $\\{8, 32, 64 \\}$ (we evaluate at $8$ since authors for SNL+ proposed to set the embedding dimension to be 2 times the dimensionality of parameters). And choose latent dimension of MAF using grid search in $\\{50, 64, 96, 128\\}$.
> >
> > We choose the initial value of the learning rate using grid search in $\\{0.01, 0.001, 0.0005\\}$.
> >
> > **Q2: Questions regarding clarity.**
> >
> > **Q2-1: ''line 21, '$\boldsymbol{\eta}$ is observation noise'...''**
> >
> > **A2-1**: In the display of line 21, we should have $\mathbf{Z} = H(\boldsymbol{\phi}; \mathbf{Z}_0) + \boldsymbol{\eta}$, where for brevity $H(\boldsymbol{\phi})$ represents running simulators for T times given initial condition $\mathbf{Z}_0$.
> >
> > **Q2-2: Question regarding clarity in ''line 113-114''.**
> >
> > **A2-2**: Given the observation $\mathbf{Z}$, Cleary et al. [2021]  trains an emulator using data simulated from a small domain of parameter space with high posterior probability. When a new set of observed data comes, most computationally expensive steps in the inference chain needs to be rerun.

---

> > > ### Author Response · Authors · 2022-08-01
> > > **Additional literature review and comparisons (P3)**
> > >
> > > **Q2-3**: Question regarding assuming $H$ is bijective at line 139.
> > >
> > > **A2-3**: The bijective assumption means that every $\mathbf{Z}_i$ is paired with a parameter $\boldsymbol{\phi}_i$ and maintains the one-on-one relationship between parameter domain and trajectory domain.
> > >
> > > **Q2-4: Question regarding providing quantification or figures of simulator $H$ error at line 174.**
> > >
> > > **A2-4**: Studies have been done for quantification of ODE errors when setting different step sizes and tolerance levels [Shampine, 2005]. However, we believe the quantification is beyond the interests of this work.
> > >
> > > **Q2-5: Question regarding abbreviation on InfoNCE.**
> > >
> > > **A2-5**: We should use Info Noise Contrastive Estimation (InfoNCE) [Oord et al., 2018].
> > >
> > > Thank you for pointing out several typos. We will correct them in the revision.
> > >
> > > **References**
> > >
> > > George Papamakarios, David Sterratt, and Iain Murray. Sequential neural likelihood: Fast likelihood-free inference with autoregressive flows. *In The 22nd International Conference on Artificial Intelligence and Statistics*, pages 837–848. PMLR, 2019
> > >
> > > Kyle Cranmer, Johann Brehmer, and Gilles Louppe. The frontier of simulation-based inference. *Proceedings of the National Academy of Sciences*, 117(48):30055–30062, 2020.
> > >
> > > Jan-Matthis Lueckmann, Jan Boelts, David Greenberg, Pedro Goncalves, and Jakob Macke. Benchmarking simulation-based inference. *In International Conference on Artificial Intelligence and Statistics*, pages 343–351. PMLR, 2021.
> > >
> > > Yanzhi Chen, Dinghuai Zhang, Michael Gutmann, Aaron Courville, and Zhanxing Zhu. *Neural approximate sufficient statistics for implicit models*. arXiv preprint arXiv:2010.10079, 2020.
> > >
> > > David Greenberg, Marcel Nonnenmacher, and Jakob Macke. Automatic posterior transformation for
> > > likelihood-free inference. *In International Conference on Machine Learning*, pages 2404–2414. PMLR, 2019.
> > >
> > > Jan-Matthis Lueckmann, Giacomo Bassetto, Theofanis Karaletsos, and Jakob H Macke. Likelihood-free inference with emulator networks. *In Symposium on Advances in Approximate Bayesian
> > > Inference*, pages 32–53. PMLR, 2019.
> > >
> > > George Papamakarios and Iain Murray. Fast $\epsilon$-free inference of simulation models with bayesian conditional density estimation. *Advances in neural information processing systems*, 29, 2016.
> > >
> > > George Papamakarios, Theo Pavlakou, and Iain Murray. Masked autoregressive flow for density
> > > estimation. *Advances in neural information processing systems*, 30, 2017.
> > >
> > > Aaron van den Oord, Yazhe Li, and Oriol Vinyals. Representation learning with contrastive predictive coding. *arXiv preprint arXiv:1807.03748*, 2018.
> > >
> > > Emmet Cleary, Alfredo Garbuno-Inigo, Shiwei Lan, Tapio Schneider, and Andrew M Stuart. Calibrate, emulate, sample. *Journal of Computational Physics*, 424:109716, 2021.
> > >
> > > Conor Durkan, Artur Bekasov, Iain Murray, and George Papamakarios. Neural spline flows. *Advances in neural information processing systems*, 32, 2019.
> > >
> > > Lawrence F Shampine. Error estimation and control for odes. *Journal of Scientific Computing*, 25(1):
> > > 3–16, 2005.
> > >
> > > Justin Alsing, Tom Charnock, Stephen Feeney, and Benjamin Wandelt. Fast likelihood-free cosmology
> > > with neural density estimators and active learning. *Monthly Notices of the Royal Astronomical
> > > Society*, 488(3):4440–4458, 2019.3689.

---

> > > > ### Comment · Reviewer_QCnJ · 2022-08-09
> > > > **Thank you for the response and score update**
> > > >
> > > > I would like to thank the authors for the diligent effort in replying to all reviews, including mine. In particular, I believe the comparison with state-of-the-art simulation-based inference methods strengthens the study, allowing to better emphasise the real contribution of the study in the context of previous literature, namely a method that can scale for costly simulators (an important and timely contribution). I am looking forward to reading the final revised version. All in all, I will revise my score from 4 to 6 and recommend acceptance of the paper.

---

### Author Response · Authors · 2022-08-08
**Any feedback on rebuttal?**

To reviewers QCnJ, fsU3, and 25cv,

Thank you again for your efforts reviewing our submission. We hope that we have adequately addressed your concerns in our rebuttal, especially the new numerical results demonstrating the empirical superiority of our Embed&Emulate method over relevant works in simulation-based inference (in response to reviewer requests). Please let us know if any further clarifications or actions from our side are needed.

Thank you,

Authors

---

> ### Author Response · Authors · 2022-08-09
> **Thank you for your feedback**
>
> Thank you again for taking the time to read our rebuttal carefully. We really appreciate your thoughtful and constructive feedback.
>
> Thank you,
>
> Authors

---

### Meta-Review · Area_Chair_dYw2 · 2022-08-24

**Recommendation:** Accept
**Confidence:** Certain

**Metareview:**

All reviewers agree that the paper proposes an interesting approach that aims at efficiently solving the inverse problem of stochastic simulators. Although some reviewers have some technical concerns at their first reviews, basically those have been resolved by the authors' responses. Thus, although there are some points that should be modified from the current form, I think we can expect the authors modify the paper in the camera-ready by reflecting the discussion. Based on these, I recommend acceptance for this paper.

**Award:**

No

---

### Decision · Program_Chairs · 2022-09-14

Accept